# MEMORY-ASSISTED SUB-PROTOTYPE MINING FOR UNIVERSAL DOMAIN ADAPTATION

**Yuxiang Lai**[1,2]    **Yi Zhou** [1,2*]   **Xinghong Liu**[1,2]    **Tao Zhou**[3]

[1] School of Computer Science and Engineering, Southeast University, China
[2] Key Laboratory of New Generation Artificial Intelligence Technology and Its Interdisciplinary Applications (Southeast University), Ministry of Education, China
[3] School of Computer Science and Engineering, Nanjing University of Science and Technology, China

## ABSTRACT

Universal domain adaptation aims to align the classes and reduce the feature gap between the same category of the source and target domains. The target private category is set as the unknown class during the adaptation process, as it is not included in the source domain. However, most existing methods overlook the intra-class structure within a category, especially in cases where there exists significant concept shift between the samples belonging to the same category. When samples with large concept shifts are forced to be pushed together, it may negatively affect the adaptation performance. Moreover, from the interpretability aspect, it is unreasonable to align visual features with significant differences, such as fighter jets and civil aircraft, into the same category. Unfortunately, due to such semantic ambiguity and annotation cost, categories are not always classified in detail, making it difficult for the model to perform precise adaptation. To address these issues, we propose a novel Memory-Assisted Sub-Prototype Mining (MemSPM) method that can learn the differences between samples belonging to the same category and mine sub-classes when there exists significant concept shift between them. By doing so, our model learns a more reasonable feature space that enhances the transferability and reflects the inherent differences among samples annotated as the same category. We evaluate the effectiveness of our MemSPM method over multiple scenarios, including UniDA, OSDA, and PDA. Our method achieves state-of-the-art performance on four benchmarks in most cases.

## 1 INTRODUCTION

Unsupervised Domain Adaptation (UDA) (Ganin and Lempitsky, 2015; Kang et al., 2019; Saito et al., 2018; Shu et al., 2018; Chen et al., 2016; Hsu et al., 2015; Kalluri et al., 2022) enables models trained on one dataset to be applied to related but different domains. Traditional UDA assumes a shared label space, limiting its applicability in diverse target distributions. Universal Domain Adaptation (UniDA) addresses these limitations by allowing the target domain to have a distinct label set. UniDA flexibly classifies target samples belonging to shared classes in the source label set, treating others as "unknown." This approach, not relying on prior knowledge about target label sets, broadens the adaptability of domain-invariant feature learning across diverse domains.

Despite being widely explored, most existing universal domain adaptation methods (Li et al., 2021; You et al., 2019; Saito and Saenko, 2021; Saito et al., 2020; Chang et al., 2022; Qu et al., 2023; Chen et al., 2022; Liang et al., 2021) overlook the internal structure intrinsically presented within each image category. These methods aim to align the common classes between the source and target domains for adaptation but usually train a model to learn the class "prototype" representing each annotated category. This is particularly controversial when significant concept shifts exist between samples belonging to the same category. These differences can lead to sub-optimal feature learning and adaptation if the intra-class structure is neglected during training. Since this type of semantic ambiguity without fine-grained category labels occurs in almost all of the DA benchmarks, all the methods will encounter this issue.

---

*Correspondence to Yi Zhou(YIZHOU.SZCN@GMAIL.COM)

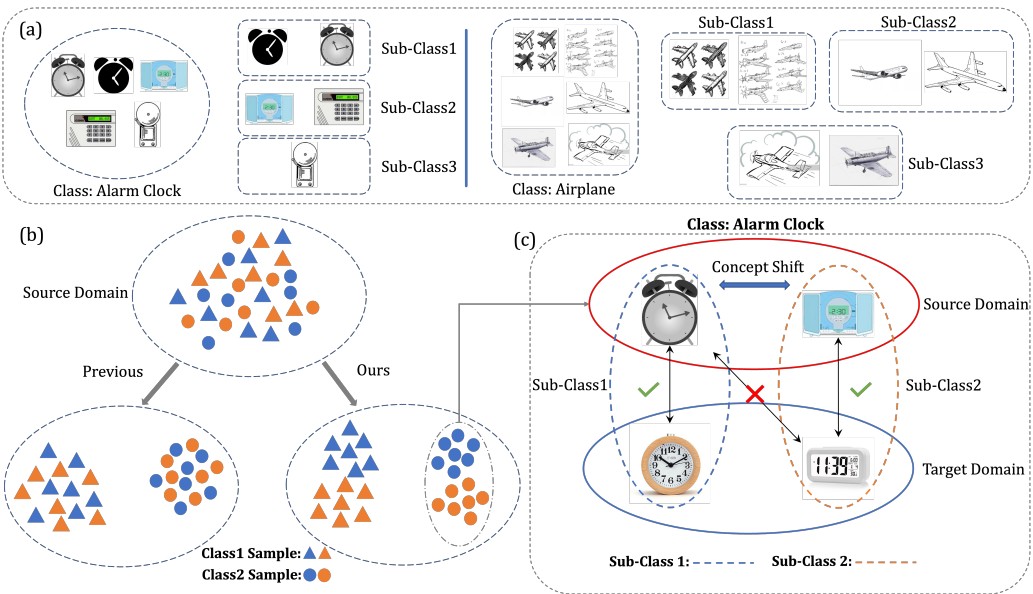

Figure 1: Illustration of our motivation. (a) Examples of concept shift and intra-class diversity in DA benchmarks. For the class of alarm clocks, we find that digital clocks, pointer clocks, and alarm bells should be set in different sub-classes. For the class of airplane, we find that images containing more than one plane, single jetliner, and turboprop aircraft should be differently treated for adaptation. (b) Previous methods utilize one-hot labels to guide classifying without considering the intra-class distinction. Consequently, the model forces all samples from the same class to converge towards a single center, disregarding the diversity in the class. Our method clusters samples with large intra-class differences into separate sub-classes, providing a more accurate representation. (c) During domain adaptation by our design, the samples in the target domain can also be aligned near the sub-class centers with similar features rather than just the class centers determined by labels.

In this paper, our objective is to propose a method that learns detailed intra-class distinctions and extracts 'sub-prototypes' to enhance alignment and adaptation. These sub-prototypes represent further subdivisions within each category-level prototype, corresponding to the 'sub-classes' of the annotated categories. Our approach revolves around employing a learnable memory structure to derive sub-prototypes for their respective sub-classes. This can optimize the construction and refinement of the feature space, bolstering the classifier's ability to distinguish class-wise relationships and improve the model's transferability across domains. As illustrated in Figure 1, the samples that are annotated as the same category often exhibit significant intra-class differences. However, previous work mainly forced them to align together for adaptation. Therefore, these methods are more likely to classify unknown classes into known classes incorrectly. Moreover, features of different sub-classes still have gaps in the feature space, making it unreasonable to align samples from distinct sub-classes, both from human perspectives and in the feature space. Aligning target domain samples at the sub-class level with source domain samples mitigates the drawback of aligning significantly different samples, making adaptation more reasonable.

Our proposed approach, named memory-assisted sub-prototype mining (MemSPM), is inspired by the memory mechanism works (Gong et al., 2019; Chen et al., 2018; Sukhbaatar et al., 2015; Rae et al., 2016). In our approach, the memory generates sub-prototypes that embody sub-classes learned from the source domain. During testing of the target samples, the encoder produces embedding that is compared to source domain sub-prototypes learned in the memory. Subsequently, an embedding for the query sample is generated through weighted sub-prototype sampling in the memory. This results in reduced domain shift before the embedding is passed to the classifier. Our proposal of mining sub-prototypes, which are learned from the source domain memory, improves the universal domain adaptation performance by promoting more refined visual concept alignment.

MemSPM approach has been evaluated on four benchmark datasets (Office-31 (Saenko et al., 2010), Office-Home (Venkateswara et al., 2017), VisDA (Peng et al., 2017), and Domain-Net (Peng et al., 2019)), under various category shift scenarios, including PDA, OSDA, and UniDA. Our MemSPM

method achieves state-of-the-art performance in most cases. Moreover, we designed a visualization module for the sub-prototype learned by our memory to demonstrate the interpretability of MemSPM. Our contributions can be highlighted as follows:

- We study UniDA problem from a new aspect, focusing on the negative impacts of ignoring intra-class structures within the same category when using one-hot labels.

- We propose Memory-Assisted Sub-Prototype Mining(MemSPM), which explores the memory mechanism to learn sub-prototypes for improving the model's adaption performance and interpretability. Meanwhile, visualizations reveal the sub-prototypes stored in memory, which demonstrate the interpretability of the MemSPM approach.

- Extensive experiments on four benchmarks verify the superior performance of our proposed MemSPM compared with previous works.

## 2 RELATED WORK

**Universal Domain Adaptation (UniDA).** You et al. (2019) proposed Universal Adaptation Network (UAN) deal with the UniDA setting that the label set of the target domain is unknown. Li et al. (2021) proposed Domain Consensus Clustering to differentiate the private classes rather than treat the unknown classes as one class. Saito and Saenko (2021) suggested that using the minimum inter-class distance in the source domain as a threshold can be an effective approach for distinguishing between "known" and "unknown" samples in the target domain. However, most existing methods (Li et al., 2021; You et al., 2019; Saito and Saenko, 2021; Saito et al., 2020; Chang et al., 2022; Qu et al., 2023; Chen et al., 2022; Liang et al., 2021; Liu et al., 2023; Zhou et al., 2022) overlook the intra-class distinction within the same category, especially there exists significant concept shift in same category.

**Concept of Prototypes.** In prior research (Kundu et al., 2022; Liu et al., 2022), prototypes have been discussed, but they differ from our MemSPM. First, in Kundu et al. (2022), subsidiary prototypes lack complete semantic knowledge and cannot address concept shifts within categories. In contrast, our sub-prototype can represent a sub-class within a category. Second, the purpose of Liu et al. (2022) is distinct from MemSPM. They aim to differentiate unknown classes. In contrast, MemSPM identifies sub-classes within a category. More details are in Appendix C.

## 3 PROPOSED METHODS

### 3.1 PRELIMINARIES

In unsupervised domain adaptation, we are provided with labeled source samples $\mathcal{D}^s = \{x_i^s, y_i^s\}_{i=1}^{n^s}$ and unlabeled target samples $\mathcal{D}^t = \{(x_i^t)\}_{i=1}^{n^t}$. As the label set for each domain in UniDA setting may not be identical, we use $C_s$ and $C_t$ to represent label sets for the two domains, respectively. Then, we denote $C = C_s \cap C_t$ as the common label set. $\hat{C}_s$, $\hat{C}_t$ are denoted as the private label sets of the source domain and target domain, respectively. We aim to train a model on $\mathcal{D}^s$ and $\mathcal{D}^t$ to classify target samples into $|C| + 1$ classes, where private samples are treated as unknown classes.

Our method aims to address the issue of intra-class concept shift that often exists within the labeled categories in most datasets, which is overlooked by previous methods. Our method enables the model to learn an adaptive feature space that better aligns fine-grained sub-class concepts, taking into account the diversity present within each category. Let $X$ denote the input query, $Z$ denote the embedding extracted by the encoder, $L$ denote the data labels, $\hat{Z}$ denotes the embedding obtained from the memory, $\hat{X}$ denote the visualization of the memory, $\hat{L}$ denotes the prediction of the input query, and the $K$ denotes the top-$K$ relevant sub-prototypes, respectively. The overall pipeline is presented in Figure 2. More details will be described in the following sub-sections.

### 3.2 INPUT-ORIENTED EMBEDDING VS. TASK-ORIENTED EMBEDDING

Typically, the image feature extracted by a visual encoder is directly used for learning downstream tasks. We call this kind of feature input-oriented embedding. However, it heavily relies on the original image content. Since different samples of the same category always vary significantly in their visual

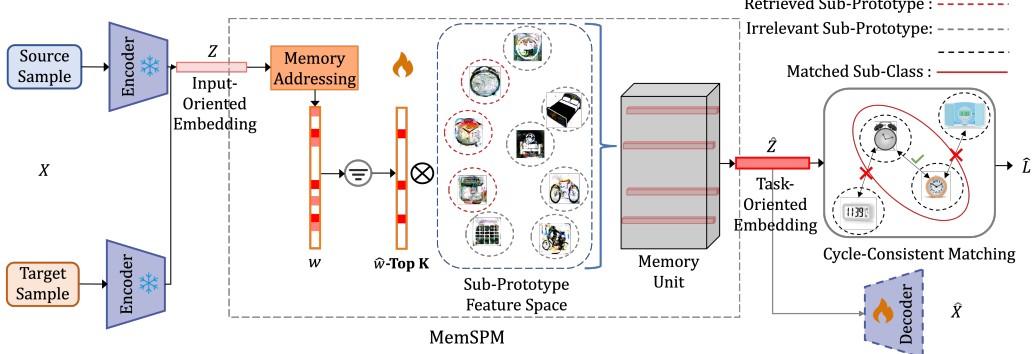

Figure 2: Our model first utilizes a fixed pre-trained model as the encoder to extract input-oriented embedding given an input sample. The extracted input-oriented embedding is then compared with sub-prototypes learned in memory to find the closest $K$. These $K$ are then weighted-averaged into a task-oriented embedding to represent the input, and used for learning downstream tasks. During the UniDA process, we adopt the cycle-consistent matching method on the task-oriented embedding $\hat{Z}$ generated from the memory. Moreover, a decoder is designed to reconstruct the image, allowing for visualizing the sub-prototypes in memory and verifying the effectiveness of sub-class learning.

features, categorization based on the input-oriented embedding is sometimes unattainable. In our pipeline, we simply adopt a CLIP-based (Radford et al., 2021) pre-trained visual encoder to extract the input-oriented embeddings, which is not directly used for learning our downstream task.

In MemSPM, we propose to generate task-oriented embedding, which is obtained by serving input-oriented embedding as a query to retrieve the sub-prototypes from our memory unit. We define $f_{encode}^{fixed}(\cdot) : X \to Z$ to represent the fixed pre-trained encoder and $f_{class}^{UniDA}(\cdot) : \hat{Z} \to \hat{L}$ to represent the UniDA classifier. The input-oriented embedding $Z$ is used to retrieve the relevant sub-prototypes from the memory. The task-oriented embedding $\hat{Z}$ is obtained using the retrieved sub-prototypes for classification tasks. In conventional ways, $\hat{Z} = Z$, which means $\hat{Z}$ is obtained directly from $Z$. Our method obtains $\hat{Z}$ by retrieving the sub-prototypes from the memory, which differentiates $\hat{Z}$ from $Z$ and reduces the domain-specific information from the target domain during testing phase. Therefore, the task-oriented information retrieved from memory will mainly have features from the source domain. Then, classifier can effectively classify, similar to how it works in source domain.

### 3.3 Memory-Assisted Sub-Prototype Mining

The memory module proposed in MemSPM consists of two key components: a memory unit responsible for learning sub-prototypes, and an attention-based addressing (Graves et al., 2014) operator to obtain better task-oriented representation $\hat{Z}$ for the query, which is more domain-invariant.

#### 3.3.1 Memory Structure with Partitioned Sub-Prototype

The memory structure in MemSPM is represented as a matrix denoted by $M \in \mathbb{R}^{N \times S \times D}$, where $N$ indicates the number of memory items stored, $S$ refers to the number of sub-prototypes partitioned within each memory item, and $D$ represents the dimension of each sub-prototype. The memory structure has learnable parameters and we use the uniform distribution to initialize memory items. For convenience, we assume $D$ is the same to the dimension of $Z \in \mathbb{R}^C$ ( $\mathbb{R}^D = \mathbb{R}^C$). Let the vector $m_{i,j}, \forall i \in [N]$ denote the $i$-th row of $M$, where $[N]$ denotes the set of integers from 1 to $N$, $\forall j \in [S]$ denote the $j$-th sub-prototype of $M$ items, where $[S]$ denotes the set of integers from 1 to $S$. Each $m_i$ denotes a memory item. Given a embedding $Z \in \mathbb{R}^D$, the memory module obtains $\hat{Z}$ through a soft addressing vector $W \in \mathbb{R}^{1 \times D}$ as follows:

$$\hat{Z}_{sn} = W \cdot M = \Sigma_d w_d \cdot m_{nsd} \text{ (Einstein summation)}, \tag{1}$$

$$w_{i,j=s_i} = \text{argmax}_j(w_{i,j}), \tag{2}$$

where $W$ is a vector with non-negative entries that indicate the maximum attention weight of each item's sub-prototype, $s_i$ denotes the index of the sub-prototype in the $i$-th item, and $w_{i,j=s_i}$ denotes the $i, j = s_i$-th entry of $W$. The hyperparameter $N$ determines the maximum capacity for memory items and the hyperparameter $S$ defines the number of sub-prototypes in each memory item. The effect of different settings of hyper-parameters is evaluated in §4.

### 3.3.2 SUB-PROTOTYPE ADDRESSING AND RETRIEVING

In MemSPM, the memory $M$ is designed to learn the sub-prototypes to represent the input-oriented embedding $Z$. We define the memory as a content addressable memory (Gong et al., 2019; Chen et al., 2018; Sukhbaatar et al., 2015; Rae et al., 2016) that allows for direct referencing of the content of the memory being matched. The sub-prototype is retrieved by attention weights $W$ which are computed based on the similarity between the sub-prototypes in the memory items and the input-oriented embedding $Z$. To calculate the weight $w_{i,j}$, we use softmax operation:

$$w_{i,j} = \frac{\exp(d(z, m_{i,j}))}{\Sigma_{n=1}^{N}\Sigma_{s=1}^{S} \exp(d(z, m_{n,s}))}, \tag{3}$$

where $d(\cdot, \cdot)$ denotes cosine similarity measurement. As indicated by Eq. 1 and 3, the memory module retrieves the sub-prototype that is most similar to $Z$ from each memory item in order to obtain the new representation embedding $\hat{Z}$. As a consequence of utilizing the adaptive threshold addressing technique(Section 3.3.3), only the top-$K$ can be utilized to obtain a task-oriented embedding $\hat{Z}$, that serves to represent the encoded embedding $Z$.

### 3.3.3 ADAPTIVE THRESHOLD TECHNIQUE FOR MORE EFFICIENT MEMORY

Limiting the number of sub-prototypes retrieved can enhance memory utilization and avoid negative impacts on unrelated sub-prototypes during model parameter updates. Despite the natural reduction in the number of selected memory items, the attention-based addressing mechanism may still lead to combining small attention-weight items into the output embedding $\hat{Z}$, which has negative impact on the classifier and sub-prototypes in memory bank. Therefore, it is necessary to impose a mandatory quantity limit on the amount of the relevant sub-prototypes retrieved. To address this issue, we apply adaptive threshold operation to restrict the number of sub-prototypes retrieved in the forward process.

$$\hat{w}_{i,j=s_i} = \begin{cases} w_{i,j=s_i}, & w_{i,j=s_i} > \lambda \\ 0, & \text{other} \end{cases} \tag{4}$$

where $\hat{w}_{i,j=s_i}$ denotes the $i, j = s_i$-th entry of $\hat{w}$, the $\lambda$ denotes the adaptive threshold:

$$\lambda = \operatorname{argmin}(topk(w_i)). \tag{5}$$

Directly implementing backward for the discontinuous function in Eq. 4 is challenging. We utilize method (Gong et al., 2019) which rewrites operation using continuous ReLU activation function as:

$$\hat{w}_{i,j=s_i} = \frac{\max(w_{i,j=s_i} - \lambda, 0) \cdot w_{i,j=s_i}}{|w_{i,j=s_i} - \lambda| + \epsilon}, \tag{6}$$

where $\max(\cdot, 0)$ is commonly referred to as the ReLU activation function, and $\epsilon$ is a small positive scalar. The prototype $\hat{Z}$ will be obtained by $\hat{Z} = \hat{W} \cdot M$. The adaptive threshold addressing encourages model to represent embedding $Z$ using fewer but more relevant sub-prototypes, leading to learning more effective features in memory and reducing the impact on irrelevant sub-prototypes.

### 3.4 VISUALIZATION AND INTERPRETABILITY

We denote $f_{decode}^{unfixed}(\cdot) : \hat{Z} \to \hat{X}$ to represent the decoder. The decoder is trained to visualize what has been learned in the memory by taking the retrieved sub-prototype as input. From an interpretability perspective, each encoded embedding $Z$ calculates the cosine similarity to find the

top-$K$ fitting sub-prototype representation for the given input-oriented embedding. Then, these sub-prototypes are combined to represent the $Z$ in $\hat{Z}$. The sub-prototype in this process can be regarded as the visual description for the input embedding $Z$. In other words, the input image is much like the sub-classes represented by these sub-prototypes. In this way, samples with significant intra-class differences will be matched to different sub-prototypes, thereby distinguishing different sub-classes. The use of a reconstruction auxiliary task can visualize the sub-prototypes in memory to confirm whether our approach has learned intra-class differences for the annotated category. The results of this visualization are demonstrated in Figure 3.

## 3.5 Cycle-Consistent Alignment and Adaption

Once the sub-prototypes are mined through memory learning, the method of cycle-consistent matching, inspired by DCC (Li et al., 2021), is employed to align the embedding $\hat{Z}$. The cycle-consistent matching is preferred due to it can provide a better fit to the memory structure compared to other UniDA methods. The other method, One-vs-All Network (OVANet), proposed by Saito et al. (Saito and Saenko, 2021), needs to train the memory multiple times, which can lead to significant computational overhead. In brief, the Cycle-Consistent Alignment provides a solution by iteratively learning a consensus set of clusters between the two domains. The consensus clusters are identified based on the similarity of the prototypes, which is measured using a similarity metric. The similarity metric is calculated on the feature representations of the prototypes. For unknown classes, we set the size $N$ of our memory during the initial phase to be larger than the number of possible sub-classes that may be learned in the source domain. This size is a hyperparameter that is adjusted based on the dataset size. Redundant sub-prototypes are invoked to represent the $\hat{Z}$, when encountering unknown classes, allowing for an improved distance separation between unknown and known classes in feature space.

**Training Objective**. The adaptation loss in our training is similar to that of DCC, as $\mathcal{L}_{DA}$:

$$\mathcal{L}_{DA} = \mathcal{L}_{ce} + \lambda_1 \mathcal{L}_{cdd} + \lambda_2 \mathcal{L}_{reg}, \tag{7}$$

where the $\mathcal{L}_{ce}$ denotes the cross-entropy loss on source samples, $\mathcal{L}_{cdd}$ denotes the domain alignment loss, and $\mathcal{L}_{reg}$ denotes the regularize (more details in Appendix E). For the auxiliary reconstruction task, we add a mean-squared-error (MSE) loss function, denoted as $\mathcal{L}_{rec}$:

$$\mathcal{L} = \mathcal{L}_{DA} + \lambda_3 \mathcal{L}_{rec} = \mathcal{L}_{ce} + \lambda_1 \mathcal{L}_{cdd} + \lambda_2 \mathcal{L}_{reg} + \lambda_3 \mathcal{L}_{rec}. \tag{8}$$

## 4 Experiments

### 4.1 Datasets and Evaluation Metrics

We first conduct the experiments in the UniDA setting (You et al., 2019) where private classes exist in both domains. Moreover, we also evaluate our approach on two other sub-cases, namely Open-Set Domain Adaptation (OSDA) and Partial Domain Adaptation (PDA).

**Datasets**. Our experiments are conducted on four datasets: Office-31 (Saenko et al., 2010), which contains 4652 images from three domains (DSLR, Amazon, and Webcam); OfficeHome (Venkateswara et al., 2017), a more difficult dataset consisting of 15500 images across 65 categories and 4 domains (Artistic images, Clip-Art images, Product images, and Real-World images); VisDA (Peng et al., 2017), a large-scale dataset with a synthetic source domain of 15K images and a real-world target domain of 5K images; and DomainNet (Peng et al., 2019), the largest domain adaptation dataset with approximately 600,000 images. Similar to previous studies (Fu et al., 2020), we evaluate our model on three subsets of DomainNet (Painting, Real, and Sketch).

As in previous work (Li et al., 2021; Saito et al., 2018; Busto et al., 2018; Cao et al., 2018; You et al., 2019), we divide the label set into three groups: common classes $C$, source-private classes $\hat{C}_s$, and target-private classes $\hat{C}_t$. The separation of classes for each of the four datasets is shown in Table 3 and is determined according to alphabetical order.

Table 3: The division on label set, Common Class ($C$) / Source-Private Class ($\hat{C}_s$) / Target Private Class ($\hat{C}_t$).

| Dataset | Class Split($C/\hat{C}_s/\hat{C}_t$) | | |
|---|---|---|---|
| | PDA | OSDA | UniDA |
| Office-31 | 10 / 21 / 0 | 10 / 0 / 11 | 10 / 10 / 11 |
| OfficeHome | 25 / 40 / 0 | 25 / 0 / 40 | 10 / 5 / 50 |
| VisDA | 6 / 6 / 0 | 6 / 0 / 6 | 6 / 3 / 3 |
| DomainNet | —— | —— | 150 / 50 / 145 |

Table 1: H-score (%) comparison in UniDA scenario on DomainNet, VisDA and Office-31,some results are cited from (Li et al., 2021; Qu et al., 2023)

| Method | Pretrain | DomainNet | | | | | | | VisDA | Office-31 | | | | | | |
|---|---|---|---|---|---|---|---|---|---|---|---|---|---|---|---|---|
| | | P2R | P2S | R2P | R2S | S2P | S2R | Avg | S2R | A2D | A2W | D2A | D2W | W2A | W2D | Avg |
| UAN (You et al., 2019) | ImageNet | 41.9 | 39.1 | 43.6 | 38.7 | 38.9 | 43.7 | 41.0 | 34.8 | 59.7 | 58.6 | 60.1 | 70.6 | 60.3 | 71.4 | 63.5 |
| CMU (Fu et al., 2020) | | 50.8 | 45.1 | 52.2 | 45.6 | 44.8 | 51.0 | 48.3 | 32.9 | 68.1 | 67.3 | 71.4 | 79.3 | 72.2 | 80.4 | 73.1 |
| DCC (Li et al., 2021) | | 56.9 | 43.7 | 50.3 | 43.3 | 44.9 | 56.2 | 49.2 | 43.0 | **88.5** | 78.5 | 70.2 | 79.3 | 75.9 | 88.6 | 80.2 |
| OVANet (Saito and Saenko, 2021) | | 56.0 | 47.1 | 51.7 | 44.9 | 47.4 | 57.2 | 50.7 | 53.1 | 85.8 | 79.4 | 80.1 | 95.4 | 84.0 | 94.3 | 86.5 |
| UMAD (Liang et al., 2021) | | 59.0 | 44.3 | 50.1 | 42.1 | 32.0 | 55.3 | 47.1 | 58.3 | 79.1 | 77.4 | 87.4 | 90.7 | **90.4** | 97.2 | 87.0 |
| GATE (Chen et al., 2022) | | 57.4 | 48.7 | 52.8 | 47.6 | 49.5 | 56.3 | 52.1 | 56.4 | 87.7 | 81.6 | 84.2 | 94.8 | 83.4 | 94.1 | 87.6 |
| UniOT (Chang et al., 2022) | | 59.3 | 47.8 | 51.8 | 46.8 | 48.3 | 58.3 | 52.0 | 57.3 | 83.7 | **85.3** | 71.4 | 91.2 | 70.9 | 90.84 | 82.2 |
| GLC (Qu et al., 2023) | | **63.3** | 50.5 | 54.9 | 50.9 | 49.6 | 61.3 | 55.1 | 73.1 | 81.5 | 84.5 | **89.8** | 90.4 | 88.4 | 92.3 | 87.8 |
| GLC Qu et al. (2023) | CLIP | 74.4 | 63.4 | 60.0 | 62.9 | 52.0 | 74.3 | 64.5 | 80.3 | 80.5 | 80.4 | 77.5 | **95.6** | 77.7 | 96.9 | 84.8 |
| DCC (Li et al., 2021) | | 61.1 | 38.8 | 51.8 | 49.3 | 49.1 | 60.3 | 52.2 | 61.2 | 82.2 | 76.9 | 83.6 | 75.2 | 85.8 | 88.7 | 82.1 |
| MemSPM+DCC | | 72.4 | 62.8 | 58.5 | 63.3 | 50.4 | 72.6 | 63.3 | **80.5** | 88.0 | 84.6 | 88.7 | 87.6 | 87.9 | 94.3 | **88.5** |

Table 2: H-score (%) comparison in UniDA scenario on Office-Home, some results are cited from (Li et al., 2021; Qu et al., 2023)

| Method | Pretrain | Office-Home | | | | | | | | | | | | |
|---|---|---|---|---|---|---|---|---|---|---|---|---|---|---|
| | | Ar2Cl | Ar2Pr | Ar2Rw | Cl2Ar | Cl2Pr | Cl2Rw | Pr2Ar | Pr2Cl | Pr2Rw | Rw2Ar | Rw2Cl | Rw2Pr | Avg |
| UAN (You et al., 2019) | ImageNet | 51.6 | 51.7 | 54.3 | 61.7 | 57.6 | 61.9 | 50.4 | 47.6 | 61.5 | 62.9 | 52.6 | 65.2 | 56.6 |
| CMU (Fu et al., 2020) | | 56.0 | 56.9 | 59.2 | 67.0 | 64.3 | 67.8 | 54.7 | 51.1 | 66.4 | 68.2 | 57.9 | 69.7 | 61.6 |
| DCC (Li et al., 2021) | | 58.0 | 54.1 | 58.0 | 74.6 | 70.6 | 77.5 | 64.3 | 73.6 | 74.9 | **81.0** | 75.1 | 80.4 | 70.2 |
| OVANet (Saito and Saenko, 2021) | | 62.8 | 75.6 | 78.6 | 70.7 | 68.8 | 75.0 | 71.3 | 58.6 | 80.5 | 76.1 | 64.1 | 78.9 | 71.8 |
| UMAD (Liang et al., 2021) | | 61.1 | 76.3 | 82.7 | 70.7 | 67.7 | 75.7 | 64.4 | 55.7 | 76.3 | 73.2 | 60.4 | 77.2 | 70.1 |
| GATE (Chen et al., 2022) | | 63.8 | 75.9 | 81.4 | 74.0 | 72.1 | 79.8 | 74.7 | 70.3 | 82.7 | 79.1 | 71.5 | 81.7 | 75.6 |
| UniOT (Chang et al., 2022) | | 67.2 | 80.5 | 86.0 | 73.5 | 77.3 | 84.3 | 75.5 | 63.3 | 86.0 | 77.8 | 65.4 | 81.9 | 76.6 |
| GLC (Qu et al., 2023) | | 64.3 | 78.2 | 89.8 | 63.1 | 81.7 | **89.1** | 77.6 | 54.2 | **88.9** | 80.7 | 54.2 | 85.9 | 75.6 |
| GLC (Qu et al., 2023) | CLIP | 79.4 | 88.9 | **90.8** | **76.3** | 84.7 | 89.0 | **71.5** | 72.9 | 85.7 | 78.2 | 79.4 | 90.0 | 82.6 |
| DCC (Li et al., 2021) | | 62.6 | 88.7 | 87.4 | 63.3 | 68.5 | 79.3 | 67.9 | 63.8 | 82.4 | 70.7 | 69.8 | 87.5 | 74.4 |
| MemSPM+DCC | | **78.1** | **90.3** | 90.7 | 81.9 | **90.5** | 88.3 | 79.2 | **77.4** | 87.8 | 78.8 | **76.2** | **91.6** | **84.2** |

**Evaluation Metrics**. We report the average results of three runs. For the PDA scenario, we calculate the classification accuracy over all target samples. The usual metrics adopted to evaluate OSDA are the average class accuracy over the known classes $OS^*$, and the accuracy of the unknown class $UNK$. In the OSDA and UniDA scenarios, we consider the balance between "known" and "unknown" categories and report the H-score (Bucci et al., 2020):

$$\text{H-score} = 2 \times \frac{OS^* \times UNK}{OS^* + UNK}, \tag{9}$$

which is the harmonic mean of the accuracy of "known" and "unknown" samples.

**Implementation Details**. Our implementation is based on PyTorch (Paszke et al., 2019). We use CLIP (Dosovitskiy et al., 2020) as the backbone pretrained by CLIP (Radford et al., 2021) for the MemSPM is hard to train with a randomly initialized encoder. The classifier consists of two fully-connected layers, which follow the previous design (Cao et al., 2018; You et al., 2019; Saito et al., 2018; Fu et al., 2020; Li et al., 2021). The weights in the $\mathcal{L}$ are empirically set as $\lambda_1 = 0.1$, $\lambda_2 = 3$ and $\lambda_3 = 0.5$ following DCC (Li et al., 2021). For a fair comparison, we also adopt CLIP as backbone for DCC (Li et al., 2021) and state-of-art method GLC (Qu et al., 2023). We use the official code of DCC (Li et al., 2021) and GLC (Qu et al., 2023) (Links in Appendix D). Regarding computational resources, MemSPM demonstrates efficient training on the Office-Home dataset using a single RTX 3090. The entire training process is completed within one day.

## 4.2 COMPARISON WITH STATE-OF-THE-ARTS

We compare our method with previous state-of-the-art algorithms in three sub-cases of unsupervised domain adaptation, namely, object-specific domain adaptation (OSDA), partial domain adaptation (PDA), and universal domain adaptation (UniDA).

**Results on UniDA**. In the most challenging setting, i.e. UniDA, our MemSPM approach achieves state-of-the-art performance. Table 7 shows the results on DomainNet, VisDA, and Office-31, and the result of Office-Home is summarized in Table 2. We mainly compare with GLC and DCC using ViT-B/16 as the backbone. On Office-31, the MemSPM+DCC outperforms the previous state-of-art method GLC by 3.7% and surpasses the DCC by 6.4%. On visda, our method surpasses the DCC

Table 4: H-score (%) comparison in OSDA scenario on Office-Home, VisDA and Office-31, some results are cited from (Li et al., 2021; Qu et al., 2023)

| Method | Pretrain | Ar2Cl | Ar2Pr | Ar2Rw | Cl2Ar | Cl2Pr | Cl2Rw | Pr2Ar | Pr2Cl | Pr2Rw | Rw2Ar | Rw2Cl | Rw2Pr | Avg | Office-31 Avg | VisDA Avg |
|---|---|---|---|---|---|---|---|---|---|---|---|---|---|---|---|---|
| OSBP (Saito et al., 2018) | | 55.1 | 65.2 | 72.9 | 64.3 | 64.7 | 70.6 | 63.2 | 53.2 | 73.9 | 66.7 | 54.5 | 72.3 | 64.7 | 83.7 | 52.3 |
| CMU (Fu et al., 2020) | | 55.0 | 57.0 | 59.0 | 59.3 | 58.2 | 60.6 | 59.2 | 51.3 | 61.2 | 61.9 | 53.5 | 55.3 | 57.6 | 65.2 | 54.2 |
| DCC (Li et al., 2021) | | 56.1 | 67.5 | 66.7 | 49.6 | 66.5 | 64.0 | 55.8 | 53.0 | 70.5 | 61.6 | 57.2 | 71.9 | 61.7 | 72.7 | 59.6 |
| OVANet (Saito and Saenko, 2021) | ImageNet | 58.6 | 66.3 | 69.9 | 62.0 | 65.2 | 68.6 | 59.8 | 53.4 | 69.3 | 68.7 | 59.6 | 66.7 | 64.0 | 91.7 | 66.1 |
| UMAD (Liang et al., 2021) | | 59.2 | 71.8 | 76.6 | 63.5 | 69.0 | 71.9 | 62.5 | 54.6 | 72.8 | 66.5 | 57.9 | 70.7 | 66.4 | 89.8 | 66.8 |
| GATE (Chen et al., 2022) | | 63.8 | 70.5 | 75.8 | 66.4 | 67.9 | 71.7 | 67.3 | 61.5 | 76.0 | 70.4 | 61.8 | 75.1 | 69.0 | 89.5 | 70.8 |
| ROS (Chang et al., 2022) | | 60.1 | 69.3 | 76.5 | 58.9 | 65.2 | 68.6 | 60.6 | 56.3 | 74.4 | 68.8 | 60.4 | 75.7 | 66.2 | 85.9 | 66.5 |
| GLC (Qu et al., 2023) | | 65.3 | 74.2 | 79.0 | 60.4 | 71.6 | 74.7 | 63.7 | 63.2 | 75.8 | 67.1 | 64.3 | 77.8 | 69.8 | 89.0 | 72.5 |
| GLC (Qu et al., 2023) | | 68.4 | 81.7 | 84.5 | **76.0** | 82.4 | **83.8** | 69.9 | 59.6 | 84.6 | **73.3** | **66.8** | 83.9 | 76.2 | 90.1 | **81.6** |
| DCC (Li et al., 2021) | CLIP | 62.9 | 73.3 | 78.4 | 49.8 | 69.2 | 75.0 | 59.3 | 61.5 | 80.9 | 68.1 | 62.5 | 80.0 | 68.4 | 81.9 | 66.2 |
| MemSPM+DCC | | **69.7** | **83.2** | **85.2** | 72.0 | 79.2 | 81.2 | **72.3** | **66.7** | **85.2** | 72.7 | 66.0 | **84.5** | **76.5** | **95.6** | 79.7 |

Table 5: H-score (%) comparison in PDA scenario on Office-Home, VisDA and Office-31, some results are cited from (Li et al., 2021; Qu et al., 2023)

| Method | Pretrain | Ar2Cl | Ar2Pr | Ar2Rw | Cl2Ar | Cl2Pr | Cl2Rw | Pr2Ar | Pr2Cl | Pr2Rw | Rw2Ar | Rw2Cl | Rw2Pr | Avg | Office-31 Avg | VisDA Avg |
|---|---|---|---|---|---|---|---|---|---|---|---|---|---|---|---|---|
| ETN (Cao et al., 2019) | | 59.2 | 77.0 | 79.5 | 62.9 | 65.7 | 75.0 | 68.3 | 55.4 | 84.4 | 75.7 | 57.7 | 84.5 | 70.4 | 96.7 | 59.8 |
| BA3US (Liang et al., 2020) | | 60.6 | **83.2** | **88.4** | 71.8 | 72.8 | 83.4 | 75.5 | 61.6 | 86.5 | 79.3 | 62.8 | **86.1** | **76.0** | **97.8** | 54.9 |
| DCC (Li et al., 2021) | | 54.2 | 47.5 | 57.5 | **83.8** | 71.6 | **86.2** | 63.7 | **65.0** | 75.2 | **85.5** | **78.2** | 82.6 | 70.9 | 93.3 | 72.4 |
| OVANet (Saito and Saenko, 2021) | ImageNet | 34.1 | 54.6 | 72.1 | 42.4 | 47.3 | 55.9 | 38.2 | 26.2 | 61.7 | 56.7 | 35.8 | 68.9 | 49.5 | 74.6 | 34.3 |
| UMAD (Liang et al., 2021) | | 51.2 | 66.5 | 79.2 | 63.1 | 62.9 | 68.2 | 63.3 | 56.4 | 75.9 | 74.5 | 55.9 | 78.3 | 66.3 | 89.5 | 68.5 |
| GATE (Chen et al., 2022) | | 55.8 | 75.9 | 85.3 | 73.6 | 70.2 | 83.0 | 72.1 | 59.5 | **84.7** | 79.6 | 63.9 | 83.8 | 74.0 | 93.7 | 75.6 |
| GLC (Qu et al., 2023) | | 55.9 | 79.0 | 87.5 | 72.5 | 71.8 | 82.7 | **74.9** | 41.7 | 82.4 | 77.3 | 60.4 | 84.3 | 72.5 | 94.1 | 76.2 |
| GLC (Qu et al., 2023) | | 63.2 | 80.7 | 86.5 | 76.0 | 77.9 | 84.1 | 74.5 | 56.8 | **84.7** | 79.8 | 57.4 | 83.0 | 75.4 | 91.5 | 86.2 |
| DCC (Li et al., 2021) | CLIP | 59.4 | 78.8 | 83.2 | 61.95 | **78.6** | 79.3 | 64.2 | 44.4 | 82.9 | 76.5 | 70.7 | 84.6 | 72.1 | 93.7 | 79.8 |
| MemSPM+DCC | | **64.7** | 81.1 | 84.5 | 74.8 | 74.7 | 77.5 | 58.7 | 60.3 | 84.2 | 70.3 | 77.2 | 85.8 | 74.5 | 94.4 | **87.9** |

by a huge margin of 16.1%. Our method also surpasses the GLC by 9.9% and the DCC by 4.5% on DomainNet. On the Office-Home, we surpass the DCC by 9.8% and the GLC by 3.7%.

**Results on OSDA and PDA**. In table 4 and table 5, we present the results on Office-Home, Office-31, and VisDA under OSDA and PDA scenarios. In the OSDA scenario, MemSPM+DCC still achieves state-of-the-art performance. Specifically, MemSPM+DCC obtains 95.6% H-score on Office-31, with an improvement of 5.5% compared to GLC and 13.7% compared to DCC. In the PDA scenario, MemSPM still achieves comparable performance compared to methods tailored for PDA. The MemSPM+DCC surpasses the DCC by 8.1% on the VisDA.

## 4.3 ABLATION STUDIES

**Visualization with Reconstruction and tSNE** We first visualize what the memory learns from Office-Home by sampling a single sub-prototype and adapting an auxiliary reconstruction task: $X \to \hat{X}$. We also provide the tSNE of the $\hat{Z}$ which retrieves the most related sub-prototypes. The visualization is shown in Figure 3. The tSNE visualization depicts the distribution of sub-classes within each category, indicative of MemSPM's successful mining of sub-prototypes. The reconstruction visualization shows what has been learned by MemSPM, demonstrating its ability to capture intra-class diversity.

**Memory-Assisted Sub-Prototype Mining (MemSPM) Impact**. As shown in Tables 7, 2, 4, and 5, MemSPM+DCC outperforms DCC across UniDA, OSDA, and PDA scenarios. MemSPM significantly enhances DCC performance with CLIP as the backbone. CLIP is applied because MemSPM's memory module, with large latent space initialized by the random normal distribution, faces challenges in retrieving diverse sub-prototypes early in training.

**Sensitivity to Hyper-parameters**. We conducted experiments on the VisDA dataset under the UniDA setting to demonstrate the impact of hyperparameters $S$ and $N$ on the performance of our method. The impact of $S$ is illustrated in Figure 3. When $S \geq 20$, the performance reaches a comparable result; for the best performance on Office-Home, $S \geq 40$ is achieved. At the same time, the performance of the model is not sensitive to the value of $N$, when $S = 30$. For parameter $K$, we conducted experiments with $K = 1$, signifying the selection of only the most relevant item. In the visualization results, the visualizations of memory items displayed meaningless images. The value of K is determined based on the attention values. During the design process, we observed that the fifth attention value is nearly zero. Consequently, we employ a top-K approach ($K = 5$) to filter out other noisy memory items. We will add these results analysis and visualizations to the revised manuscript.

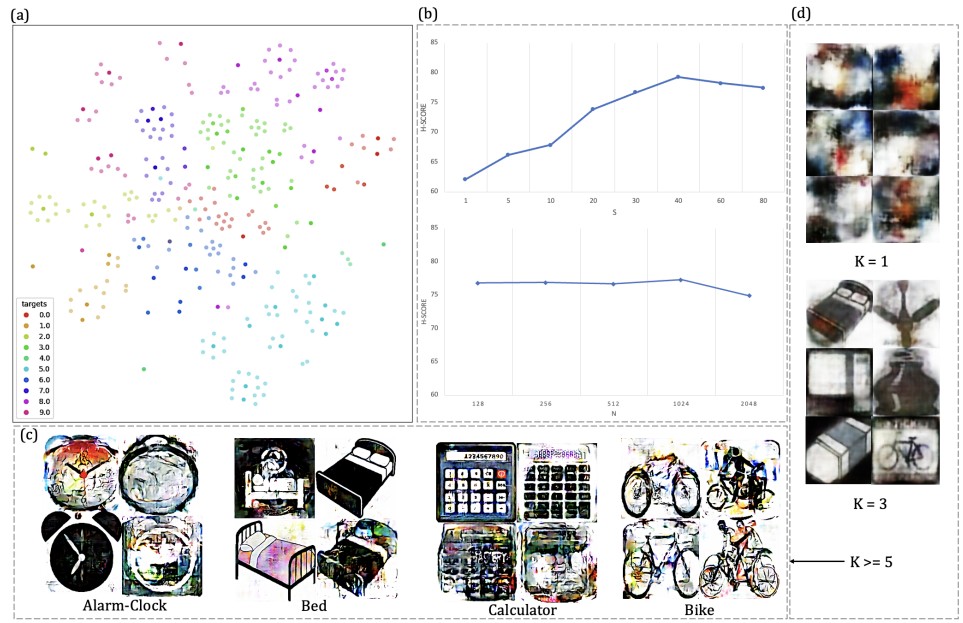

Figure 3: (a) The tSNE visualization shows the feature space of the sub-classes belonging to each category, which demonstrates the MemSPM mining the sub-prototypes successfully. (b) The results of different values of $S$ and $N$. (c) The reconstruction visualization shows what has been learned in the memory, which demonstrates the intra-class diversity has been learned by MemSPM. (d) The visualization of varying $K$ shows that insufficient values hinder the learning of appearance features.

Table 6: Ablation Studies

| Method | Pretrain | Ar2Cl | Ar2Pr | Ar2Rw | Cl2Ar | Cl2Pr | Cl2Rw | Pr2Ar | Pr2Cl | Pr2Rw | Rw2Ar | Rw2Cl | Rw2Pr | Avg |
|---|---|---|---|---|---|---|---|---|---|---|---|---|---|---|
| CLIP-Baseline | CLIP | 64.6 | 84.3 | 78.1 | 73.7 | 88.2 | 86.5 | 68.1 | 68.7 | **89.6** | 68.5 | 69.4 | 86.6 | 77.2 |
| DCC+MemSPM | ImageNet | 57.1 | 85.0 | 88.4 | 60.8 | 61.1 | 85.2 | **83.5** | 76.1 | 87.5 | **82.7** | **77.3** | 76.4 | 76.7 |
| DCC+MemSPM | CLIP | **78.1** | **90.3** | **90.7** | **81.9** | **90.5** | **88.3** | 79.2 | 77.4 | 87.8 | 78.8 | 76.2 | **91.6** | **84.2** |
| DCC+MemSPM | None | 50.7 | 78.4 | 85.6 | 50.2 | 60.7 | 67.1 | 58.2 | 44.1 | 77.9 | 67.1 | 50.3 | 81.7 | 64.33 |
| Fixed Threshold=0.005 DCC+MemSPM | CLIP | 64.6 | 86.7 | 87.4 | 63.3 | 68.5 | 79.3 | 65.9 | 65.8 | 81.4 | 70.7 | 68.8 | 85.5 | 73.9 |
| DCC+MemSPM Without $Lcdd$ | CLIP | 75.9 | 75.4 | 86.4 | 80.1 | 71.6 | 87.5 | 70.1 | **87.1** | 88.7 | 74.2 | 73.5 | 88.8 | 79.8 |

**Effect of CLIP-based Feature**. As shown in Table 6, we have conducted experiments to compare ViT-B/16 (pre-trained by CLIP), ViT-B/16 (pre-trained on ImageNet), and ViT-B/16 (without pre-training). The performance of MemSPM on Officehome using ViT-B/16 (ImageNet) is 76.7% (H-score), which is 7.5% lower than MemSPM using ViT-B/16 (pre-trained on CLIP). Additionally, the ViT-B/16 (without pre-training) only achieves 64.3%, which is 19.9% lower than that using ViT-B/16 (pre-trained on CLIP).

**Effect of Adaptive Threshold** As shown in Table 6, to demonstrate the effectiveness of the adaptive threshold, we find a best-performed fixed threshold of 0.005 through experiments. It limits the memory to learn sub-prototypes, which only achieved 73.9% (H-score) on Officehome.

**Effect of Loss** As shown in Table 6, we experimented with loss contributions. $\mathcal{L}_{ce}$ for classification is essential; removing $\mathcal{L}_{cdd}$ led to a 4.4% drop (79.8%). Optimal coefficients for $\mathcal{L}_{ce}$ ($\lambda_1 = 0.1$) and $\mathcal{L}_{cdd}$ ($\lambda_2 = 3$) achieves the best performance. The reconstruction loss ($\mathcal{L}_{rec}$) slightly improved performance, mainly for visualizing sub-prototypes.

## 5 CONCLUSION

In this paper, we propose the Memory-Assisted Sub-Prototype Mining (MemSPM) method, which can learn the intra-class diversity by mining the sub-prototypes to represent the sub-classes. Compared with previous methods, which overlook the intra-class structure by using the one-hot label, our MemSPM can learn the class feature from a more subdivided sub-class perspective to improve adaptation performance. At the same time, the visualization of the tSNE and reconstruction demonstrates the sub-prototypes have been well learned as we expected. Our MemSPM method exhibits superior performance in most cases compared with previous state-of-the-art methods on four benchmarks.

ACKNOWLEDGEMENT

This work was partially supported by the National Natural Science Foundation of China (Grants No 62106043, 62172228), and the Natural Science Foundation of Jiangsu Province (Grants No BK20210225).

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

# Appendix

In the supplementary material, we provide additional visualization results, limitations, potential negative societal impacts and compute requirements of the MemSPM. In the pursuit of reproducible research, we will make the demo and network weights of our code available to the public.

This supplementary is organized as follows:

## A  NOTATIONS

Table 7:

| | Symbol | Description |
|---|---|---|
| Model | $f_{encode}^{fixed}(\cdot)$ | Fixed image encoder |
| | $f_{decode}^{unfixed}(\cdot)$ | Unfixed reconstruction decoder |
| | $f_{class}^{UniDA}$ | UniDA classifier |
| | $M$ | Memory unit |
| | $W$ | Weight vector |
| Space | $\mathcal{D}^s$ | Labeled source dataset |
| | $\mathcal{D}^t$ | Unlabeled target dataset |
| | $C$ | Common label set |
| | $C_s$ | Source label set |
| | $C_t$ | Target label set |
| | $\hat{C}_s$ | Source private label set |
| | $\hat{C}_t$ | Target private label set |
| Samples | $X$ | Input image |
| | $\hat{X}$ | Reconstruction of image |
| | $Z$ | Input-oriented embedding |
| | $\hat{Z}$ | Task-oriented embedding |
| | $L$ | Label of the image |
| | $\hat{L}$ | Prediction of image |
| Measures | $w_{i,j}$ | Attention weight measurement between $Z$ and sub-prototype |
| | $d(\cdot, \cdot)$ | Cosine similarity measurement |
| | $\hat{w_{i,j}}$ | Adaptive threshold operation on $w_{i,j}$ |
| Hyperparameters | $N$ | Number of memory items |
| | $S$ | Number of sub-prototypes partitioned in each memory item |
| | $D$ | Dimension of each sub-prototype |
| | $K$ | Top-K relevant sub-prototypes of $Z$ |

## B  LIMITATION

Training the memory unit of MemSPM is challenging when adopting the commonly used ResNet-50 as the backbone. This is due to the memory unit's composition of massive randomly initialized tensors. During the early stage of training, there is a lack of discriminability in the input-oriented embedding, which leads to addressing only a few sub-prototypes. This decoupling of the memory unit from the input data necessitates using a better pre-trained model (ViT-B/16 pre-trained on CLIP) and fixing the encoder to reduce computation requirements. Additionally, the number of sub-prototypes in one memory item might need to be adjusted for the diversity of the category.

## C  COMPARSION BETWEEN RELATED PROTOTYPE CONCEPTS

The related concept of the prototype is mentioned in some previous works Kundu et al. (2022); Liu et al. (2022), there are clear differences between theirs and our MemSPM.

First, the meaning of prototype is different between Kundu et al. (2022) and ours. In the Kundu et al. (2022), the subsidiary prototype is extracted from randomly cropped images, which means the subsidiary prototypes only represent the low-level, morphological, and partial features of the image. These subsidiary prototypes don't have complete semantic knowledge, and the method can't learn the concept shift in the same category. Moreover, they still used the labeled category directly for alignment and adaptation. These prototypes can't represent some part of the samples in one category.

In contrast, the MemSPM allows memory items to extract complete semantic knowledge and maintain domain-invariant knowledge. To accomplish this, we use input-oriented embedding, which involves comparing the entire image feature with memory items. The memory can then sample a task-oriented embedding that represents the semantic knowledge of the input-oriented embedding. Our approach is designed to obtain a domain-invariant and semantic feature for categories with significant domain shifts. As a result, each sub-prototype can represent a sub-class in one category.

Second, the purpose of Liu et al. (2022) is very different from our MemSPM. They aim to learn differences among unknown classes, which is like the DCC. It still extracts features and aligns the class across different domains directly based on one-hot labels, and is not concerned with the concept shift and difference in one category. However, our method can mine the sub-classes in one category when there exist significant concept shifts, reflecting the inherent differences among samples annotated as the same category. This helps universal adaptation with a more fine-grained alignment or to make significant decisions without human supervision.

## D  IMPLEMENTATION DETAILS

**DCC.** We use ViT-B/16 (Dosovitskiy et al., 2020) as the backbone. The classifier is made up of two FC layers. We use Nesterov momentum SGD to optimize the model, which has a momentum of $0.9$ and a weight decay of 5e-4. The learning rate decreases by a factor of $(1 + \alpha \frac{i}{N})^{-\beta}$, where $i$ and $N$ represent current and global iteration, respectively, and we set $\alpha = 10$ and $\beta = 0.75$. We use a batch size of 36 and the initial learning rate is set as 1e-4 for Office-31, and 1e-3 for Office-Home and DomainNet. We use the settings detailed in Li et al. (2021). PyTorch Paszke et al. (2019) is used for implementation.

**GLC.** We use ViT-B/16 (Dosovitskiy et al., 2020) as the backbone. The SGD optimizer with a momentum of $0.9$ is used during the target model adaptation phase of GLC (Qu et al., 2023). The initial learning rate is set to 1e-3 for Office-Home and 1e-4 for both VisDA and DomainNet. The hyperparameter $\rho$ is fixed at $0.75$ and $|L|$ at 4 across all datasets, while $\eta$ is set to $0.3$ for VisDA and $1.5$ for Office-Home and DomainNet, which corresponds to the settings detailed in (Qu et al., 2023). PyTorch (Paszke et al., 2019) is used for implementation.

**Existing code used.**

- DCC (Li et al., 2021):
  https://github.com/Solacex/Domain-Consensus-Clustering
- GLC (Qu et al., 2023):
  https://github.com/ispc-lab/GLC

- PyTorch (Paszke et al., 2019):
  https://pytorch.org/

**Existing datasets used.**

- Office-31 (Saenko et al., 2010):
  https://www.cc.gatech.edu/âĹijjudy/domainadapt
- Office-Home (Venkateswara et al., 2017):
  https://www.hemanthdv.org/officeHomeDataset.html
- DomainNet (Peng et al., 2019):
  http://ai.bu.edu/M3SDA
- VisDA (Peng et al., 2017):
  http://ai.bu.edu/visda-2017/

**Compute Requirements.** For our experiments, we used a local desktop machine with an Intel Core i5-12490f, a single Nvidia RTX-3090 GPU, and 32GB of RAM. When we adapt the batch-size used in DCC (Li et al., 2021), our MemSPM only occupies 4GB of GPU memory during training as the result of fixing the encoder.

## E    DETAILS OF DOMAIN CONSENSUS CLUSTERING

**Domain Consensus Clustering (DCC).** They leverage Contrastive Domain Discrepancy (CDD) to facilitate the alignment over identified common samples in a class-aware style. They impose $L_{\text{CDD}}$ to minimize the intra-class discrepancies and enlarge the inter-class gap. Consequently, the enhanced discriminability, in turn, enables DCC to perform more accurate clustering. Details of CDD are provided in: https://openaccess.thecvf.com/content/CVPR2021/supplemental/Li_Domain_Consensus_Clustering_CVPR_2021_supplemental.pdf.

## F    DISCUSSION OF MOTIVATION

Illustrated in Figure 1, our motivation arises from the recognition that samples annotated within the same category often exhibit significant intra-class differences and concept shifts. In Figure 1 (a), we visually depict this phenomenon, showcasing samples labeled as the class "alarm clock" further divided into three distinct sub-classes: desktop alarm clock, digital alarm clock, and wall alarm clock. Similarly, the class "airplane" encompasses multiple sub-classes, such as propeller planes and jet aircraft. This demonstrates that the semantic ambiguity and annotation cost lead to the inefficiency of class alignment. Previous methods often force samples with significant concept shifts to align together during adaptation, increasing the likelihood of misclassifying unknown classes into known classes. In contrast, our proposed method addresses this challenge by introducing sub-prototypes, refining the features of known classes by separating them into sub-classes, and reducing the risk of negative transfer.

## G    VISUALIZATION

We provide more results of visualization in Figure 4 and Figure 5 to reveal sub-prototypes stored in the memory unit, which demonstrates that our MemSPM approach can learn the intra-class concept shift.

## H    POTENTIAL SOCIETAL IMPACT

Our finding of the intra-class concept shift may influence future work on domain adaption or other tasks. They can optimize the construction and refinement of the feature space by considering the intra-class distinction. The MemSPM also provides a method that can be used to demonstrate the interpretability of the model for further deployment. However, the utilization of the MemSPM method for illegal purposes may be facilitated by its increased availability to organizations or individuals.

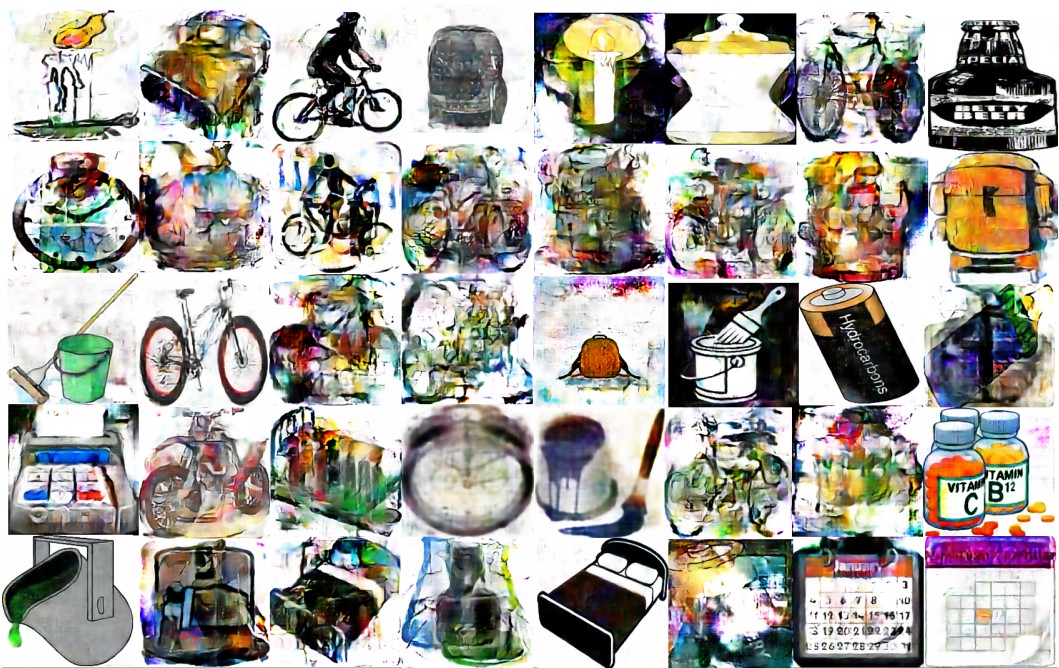

Figure 4: The reconstruction visualization shows what has been learned in the memory, which demonstrates the intra-class diversity has been learned by MemSPM.

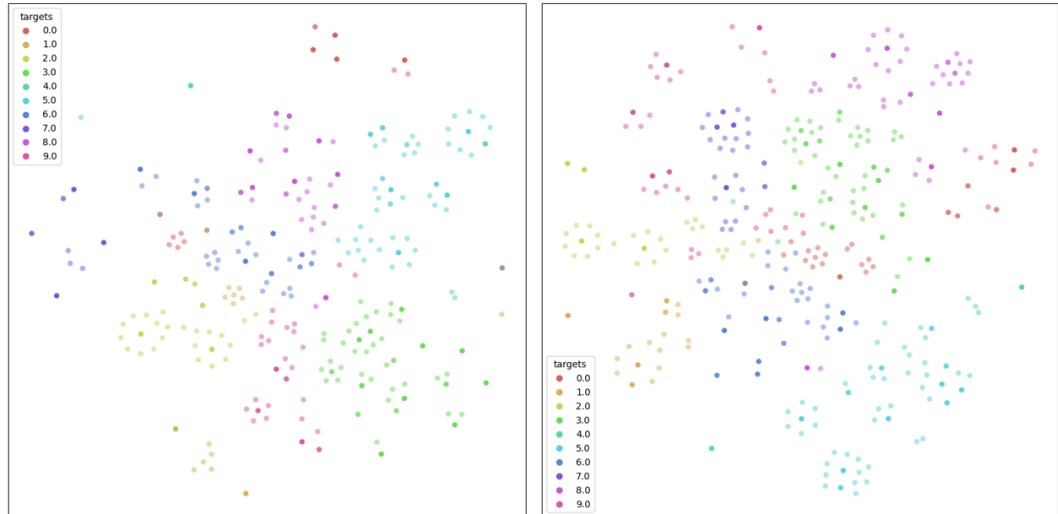

Figure 5: The tSNE visualization shows the distribution of the retrieved sub-prototypes and demonstrates that the sub-classes have been learned by MemSPM.

The MemSPM method may be susceptible to adversarial attacks as all contemporary deep learning systems. Although we demonstrate increased performance and interpretability compared to the state-of-the-art methods, negative transfer is still possible in extreme cases of domain shift or category shift. Therefore, our technique should not be employed in critical applications or to make significant decisions without human supervision.

