# OpenReview forum: "Memory-Assisted Sub-Prototype Mining for Universal Domain Adaptation"
_ICLR.cc/2024/Conference — ICLR 2024 poster_

### Official Review · Reviewer_dY2C · 2023-10-30

**Soundness:** 3 good
**Presentation:** 2 fair
**Contribution:** 2 fair
**Rating:** 6
**Confidence:** 4

**Summary:**

This paper introduces the Memory-Assisted Sub-Prototype Mining (MemSPM) method for Universal Domain Adaptation. The primary goal is to better align classes and reduce feature gaps between source and target domains, addressing the limitations of most existing models that do not consider intra-class structures, especially when significant concept shifts exist within the same category. The proposed MemSPM enhances model performance by effectively identifying and learning from sub-classes that have considerable concept shifts, leading to a more reasonable feature space, improved transferability, and interpretability. Experimental results across several scenarios (UniDA, OSDA, PDA) show that the MemSPM method outperforms existing benchmarks in many instances.

**Strengths:**

The Memory-Assisted Sub-Prototype Mining (MemSPM) brings a novel perspective to the Universal Domain Adaptation domain by addressing the challenge of significant concept shifts within the same category. The paper highlights the limitation of existing models and provides a robust solution to enhance transferability and reflect inherent sample differences. Experimental results support the claims and showcase the model's superior performance in many scenarios.

**Weaknesses:**

While the paper introduces a novel approach, it could benefit from clearer explanations and visual aids, making the methodology more accessible to readers. Additionally, the paper could delve deeper into potential drawbacks or limitations of the proposed method. Comparison with a broader set of benchmarks might also provide a more comprehensive understanding of the model's applicability and robustness.

**Questions:**

- How does the Memory-Assisted mechanism integrate with existing architectures, and what is its computational overhead?

- Are there any specific scenarios or datasets where the MemSPM method might not be as effective?

- Can the authors provide more insights into the annotation/training costs mentioned and how the proposed method addresses this?

---

> ### Author Response · Authors · 2023-11-21
>
> **Re_Q1:** How does the Memory-Assisted mechanism integrate with existing architectures, and what is its computational overhead?
>
> The integration of the MemSPM is straightforward, requiring a simple insertion between the encoder and classifier within existing architectures. As for computational overhead, the training of MemSPM is efficiently conducted on OfficeHome using a single RTX 3090 within a day, making it a computationally affordable and practical solution.
>
> **Re_Q2:**  Are there any specific scenarios or datasets where the MemSPM method might not be as effective?
>
> Our motivation primarily addresses the challenges posed by concept shifts within labeled classes. Consequently, the effectiveness of the MemSPM method may be limited in scenarios where datasets exhibit minimal concept shifts and are meticulously annotated. However, it's crucial to note that the MemSPM method has demonstrated notable efficacy in addressing significant concept shifts, as observed in commonly used domain adaptation datasets. For instance, in four existing widely utilized DA datasets, approximately 90% of categories exhibit substantial concept shifts, as depicted in Figure 1(a). Moreover, larger datasets like ImageNet showcase even more pronounced intra-class distinctions, further validating the applicability and effectiveness of the MemSPM method across diverse datasets with substantial concept shifts. Therefore, the common existence of intra-class concept shift is worthy of consideration and necessitates resolution in real-world applications.
>
> **Re_Q3:** Can the authors provide more insights into the annotation/training costs mentioned and how the proposed method addresses this?
>
> **Annotation cost:** Illustrated in Figure 1, our motivation arises from the recognition that samples annotated within the same category often exhibit significant intra-class differences and concept shifts. In Figure 1 (a), we visually depict this phenomenon, showcasing samples labeled as the class "alarm clock" further divided into three distinct sub-classes: desktop alarm clock, digital alarm clock, and wall alarm clock. Similarly, the class "airplane" encompasses multiple sub-classes, such as propeller planes and jet aircraft. This demonstrates that the semantic ambiguity and annotation cost lead to the inefficiency of class alignment. Previous methods often force samples with significant concept shifts to align together during adaptation, increasing the likelihood of misclassifying unknown classes into known classes. In contrast, our proposed method addresses this challenge by introducing sub-prototypes, refining the features of known classes by separating them into sub-classes and reducing the risk of negative transfer.
>
> **Training cost:** Utilizing CLIP as the encoder, we acknowledge its challenges in training and fine-tuning. Therefore, we employ a fixed CLIP encoder to mitigate the training cost. As mentioned in response to Q1, our method can operate efficiently, requiring only one RTX 3090 GPU within a single day.

---

### Official Review · Reviewer_mKia · 2023-10-31

**Soundness:** 3 good
**Presentation:** 2 fair
**Contribution:** 2 fair
**Rating:** 5
**Confidence:** 4

**Summary:**

This paper proposes a Memory-Assisted Sub-Prototype Mining (MemSPM) for the UniDA problem, which involves the idea of sub-prototypes.

**Strengths:**

The idea of sub-prototype is reasonable and worth trying for UniDA.

**Weaknesses:**

1. The comparison of DCC and MemSPM+DCC is not fair, due to the usage of CLIP in MemSPM. The author SHOULD replace the CLIP with a learnable encoder to achieve a fair comparison.
2. Only involving the idea of sub-prototype is not novel enough for the acceptance of top conference. How to use it to solve the concept shift in Fig.1 for UniDA is the key. However, in my opinion, the loss in MemSPM (i.e. Cycle Consistent Matching) is exactly the same with DCC. So, I can not find anything new here.

**Questions:**

See the weakness.

---

> ### Author Response · Authors · 2023-11-21
>
> **Re_Q1:**
>
> Although CLIP-based embedding does have some cross-domain knowledge, it still cannot address the large domain gap that existed in the benchmarks. The baseline that simply adopts the CLIP encoder has been tested on the OfficeHome dataset only achieving **77.2%** (H-score), which is **7.0%** lower than our MemSPM.
>
> In **Table 2** and **Table 3**, all methods, including GLC, DCC, and MemSPM, utilize the ViT-B/16 backbone (pre-trained by CLIP), ensuring a consistent and fair comparison. It's crucial to note that GLC represents the state-of-the-art method in UniDA, and DCC serves as the baseline for our approach.
>
> **Re_Q2:**
>
> Unlike previous methods that heavily relied on aligning samples based on labeled categories, our research reveals the inadequacy of addressing current benchmarks solely through label-based alignment. Consequently, the introduction of sub-prototypes becomes paramount, providing a crucial means to differentiate between common and private visual appearances. It's crucial to note that this differentiation may not align strictly with the conventional common/private concepts based on labeled categories.
>
> Beyond outlining our motivation, we meticulously designed, experimentally validated and successfully fulfilled the objectives of our proposed method. Our method offers satisfactory visualizations, demonstrating a certain level of interpretability. It is essential to assert the necessity and distinctive features of several design modules in our approach, such as the input-oriented, task-oriented, memory structure with partitioned sub-prototype, adaptive threshold technique for more efficient memory, and others. Here, we delve into the highlights of our approach, underscoring the non-trivial nature of sub-prototypes.
>
> Our method effectively mitigates concept shifts by aligning target domain samples at the sub-prototype level with source domain samples. This strategic alignment addresses the limitations of previous methods that forced dissimilar samples to align, reducing the risk of misclassifying unknown classes into known classes. In the feature space, samples’ features of different sub-classes still have gaps in the feature space, so it is not reasonable to align samples from different sub-prototypes together not only in human understanding but also in the learned feature space.
>
> The retrieval pattern of sub-prototypes relies on comparing the similarity of input-oriented embeddings with memory items. For common classes, our method identifies similar sub-prototypes from the learned memory, creating task-oriented embeddings for downstream classification. However, private classes receive redundant memory items not utilized by the source domain, resulting in task-oriented embeddings distinct from those of common classes.
>
> Moreover, MemSPM leverages input-oriented embeddings that encompass the entire image feature, enabling the memory to extract comprehensive semantic knowledge and maintain domain-invariant information. This approach ensures a more accurate representation of each sub-class in one category, addressing significant domain shifts and contributing to domain-invariant and semantic features.
>
> We would like to emphasize that while Cycle Consistent Matching serves as a complementary mechanism to our method for addressing UniDA, its inclusion does not diminish the unique contributions and novelty of our approach to solving these issues.
>
> For further clarification and conviction, we will enhance the explanation of how our method alleviates concept shift in the revised paper.

---

### Official Review · Reviewer_S3ot · 2023-11-01

**Soundness:** 2 fair
**Presentation:** 1 poor
**Contribution:** 2 fair
**Rating:** 3
**Confidence:** 4

**Summary:**

In this study, the authors introduce the Memory-Assisted Sub-Prototype Mining method, which emphasizes the significance of the internal structure within the category. Unlike conventional methods that treat the classes as a whole, MemSPM refines class features and mines sub-prototypes to represent sub-classes, thereby enhancing adaptation performance. The study provides extensive experiments on four benchmarks to validate its performance.

**Strengths:**

1.	This paper proposes a novel method called Memory-Assisted Sub-Prototype Mining, designed to significantly improve the model’s adaptability. Additionally, the approach provides insightful visualizations, providing a clear understanding of the methodology.
2.	The effectiveness of the proposed method has been assessed through extensive experiments.

**Weaknesses:**

1.	While the proposed method demonstrates effectiveness, it appears that the learned sub-prototype does not effectively handle the most challenging aspect of UniDA, such as identifying common and private samples. The Sub-Prototype Mining method seems better suited for general domain adaptation problems rather than specifically addressing the complexities of UniDA.
2.	The paper lacks explicit clarification on how the sub-prototype is initialized and updated, leaving a gap in understanding the methodology.
3.	Section 3.5 is unclear and requires further elaboration. The concept of cycle-consistent alignment needs to be more explicitly reflected in the training phase. Additionally, a comparison of performance differences with other alignment strategies would enhance the paper's comprehensiveness.
4.	A significant portion of the paper relies heavily on DCC, and the proposed method appears to primarily involve weighting input-oriented embeddings to obtain task-oriented embeddings, which seems to be the only deviation from DCC. This raises concerns about the novelty and originality of the proposed approach.

**Questions:**

1.  In part 3.3.1, there are N memory items stored. However, the explanation of how these N items are produced and the relationship between them is not clear. Additionally, this paper does not provide a clear explanation of how the sub-prototypes are generated and updated.

2.  In Table 6, why only show the performance without the domain alignment loss?  Need to report the performance without the regularized loss and the reconstruction loss.

---

> ### Author Response · Authors · 2023-11-21
>
> **Re_Q1:**
>
> The most challenging aspect of UniDA lies in effectively distinguishing between known and unknown classes, aiming for more accurate domain adaptation, and preventing negative transfer. As depicted in our motivation (see **Figure 1**), samples annotated under one category often exhibit significant intra-class variations. In previous approaches, attempts were made to force the alignment of these samples for adaptation, making them more likely to incorrectly classify unknown classes into known classes. Therefore, the key emphasis lies in performing a more detailed cross-domain alignment.
>
> Unlike previous methods that relied on aligning samples based on labeled categories, our research indicates that the existing benchmarks and even real scenarios cannot be sufficiently addressed through alignment solely based on labels. Consequently, the introduction of sub-prototypes becomes crucial, enabling the differentiation between common and private visual appearances. It's important to note that this differentiation may not necessarily align with the concepts of common/private based on labeled categories.
>
> Furthermore, the issue of concept shift persists in unknown classes. By employing sub-prototypes, our method not only refines the features of known classes by separating sub-classes but also refines the features of sub-classes within unknown classes. For common classes, the method identifies similar sub-prototypes from the learned memory, creating task-oriented embeddings for downstream classification. However, in the case of private classes, redundant memory items are obtained, which have not been used by the source domain. Consequently, their task-oriented embedding differs significantly from common classes, effectively reducing negative transfer between known and unknown classes.
>
> Our approach exhibits more pronounced advantages in the UniDA setting, where aligning category concepts imposes higher demands. We sincerely thank the reviewers for their valuable suggestions and commit to extending our method to broader research on general domain adaptation tasks.
>
> **Re_Q2:**
>
> Thanks for the advice. The memory structure has learnable parameters and we use the uniform distribution to initialize memory items. We will add these to the revised manuscript.
>
> The memory learns sub-prototypes that embody sub-classes learned from the source domain. During testing of the target samples, the encoder produces embedding that is compared to source domain sub-prototypes learned in the memory. Subsequently, an embedding for the query sample is generated through weighted sub-prototype sampling in the memory. This results in reduced domain shifts before the embedding gives into the classifier.
>
> **Re_Q3:**
>
> The Cycle-Consistent Alignment is used to match the target domain sub-prototype clusters to the source domain and then align two similar sub-classes together. Due to space constraints, we did not describe the details of this module, we will make the clarification clearer in the revised manuscript.
>
> The MemSPM indeed can be used in other methods, but we chose the DCC that most fit our motivation in the UniDA task. The experiments listed in **Tables 2,3,4, and 5** have proved the effectiveness of our MemSPM. Thanks for the advice. We will apply the MemSPM to other methods in the revision.
>
> **Re_Q4:**
>
>
> In **Table 2** and **Table 3**, we adopted the ViT-B/16 backbone (pre-trained by CLIP) consistently across GLC, DCC, and MemSPM, ensuring a fair comparison. GLC represents the state-of-the-art (SOTA) method in UniDA, while DCC serves as the basis for our approach. The experiments robustly demonstrate the effectiveness of our design, with MemSPM significantly outperforming the DCC method.
>
> Our MemSPM goes beyond a simple process of weighting input-oriented embeddings to obtain task-oriented embeddings. It offers compelling visualizations, showcasing a noteworthy level of interpretability. It's imperative to highlight the necessity and distinctive features of various design modules in our approach, including input-oriented and task-oriented components, the memory structure with partitioned sub-prototypes, and adaptive threshold techniques for more efficient memory, among others. Here, we delve into the intricacies of these design aspects, underscoring the non-trivial nature of sub-prototypes.
>
> Moreover, the challenge of concept shift persists in unknown classes. Through the utilization of sub-prototypes, our method not only refines the features of known classes by separating sub-classes but also enhances the features of sub-classes within unknown classes. While the method identifies similar sub-prototypes from the learned memory for common classes, it acquires redundant memory items for private classes, previously unused by the source domain. As a result, the task-oriented embedding for private classes significantly differs from common classes, effectively mitigating negative transfer between known and unknown classes.

---

> > ### Comment · Reviewer_S3ot · 2023-11-22
> >
> > Thanks for the response. The author addresses some of my concerns, but I still have reservations regarding the novelty of the paper. I share similar concerns with reviewer mKia; primarily, the idea of the sub-prototype is not novel enough, and the methods heavily rely on DCC. Therefore, I maintain my original score.

---

### Official Review · Reviewer_59Jy · 2023-11-03

**Soundness:** 3 good
**Presentation:** 3 good
**Contribution:** 3 good
**Rating:** 8
**Confidence:** 5

**Summary:**

This paper focuses on Universal Domain Adaptation (UniDA), a practical DA setting that does not make any assumptions on the relation between source and target label sets. The goal is to adapt a classifier from source to target domain such that both source and target domains may have their own private classes apart from shared classes. The paper claims that existing UniDA methods overlook the intrinsic structure in the categories, which leads to suboptimal feature learning and adaptation. Hence, they propose memory-assisted sub-prototype mining (MemSPM) that learns sub-prototypes in a memory mechanism to embody the subclasses from the source data. Then, for target samples, weighted sub-prototype sampling is used before passing the embedding to a classifier, which results in reduced domain shift for the embedding. They also propose an adaptive thresholding technique to select relevant sub-prototypes. Finally, they adopt the cycle consistent matching loss objective from DCC [24] along with an auxiliary reconstruction loss for training. They show results on UniDA, Partial DA, and Open-Set DA using standard benchmarks like Office-31, Office-Home, VisDA, and DomainNet.

**Strengths:**

* The motivating ideas for the approach are interesting and intuitive. Further, the technical contributions are novel as well as effective.

* It is intriguing that the auxiliary reconstruction task provides interpretability, which is usually not possible in existing DA solutions.

* The paper is fairly well-written and easy to understand.

* With their method and the advantages of a CLIP-pretrained ViT model, they achieve large improvements over existing ResNet-based methods. While they also show improvements over some existing methods using the CLIP-pretrained model, this will serve as a new strong baseline for future UniDA work.

**Weaknesses:**

* Missing sensitivity analysis to hyperparameter $K$ in adaptive thresholding
    * There is no information on how $K$ is chosen to be used in the top-$K$ operation used to compute the adaptive threshold $\lambda$ in Eq. 5. It may be difficult to tune for new datasets.
    * As per Table 6, using a fixed threshold drops the performance by almost 10%, which is alarming. Hence, the sensitivity to $K$ could be important and needs to be studied.
    * One simple baseline to get rid of $K$ could be to use a higher temperature in the attention computation, i.e. to make the attention distribution sharper.

* Incomplete analysis of sensitivity to hyperparameter $S$
    * As per Fig. 3b, the performance improves drastically when $S$ is increased. But the analysis is only up to $S=40$ and it seems that the performance should improve further if $S$ is increased more.
    * The paper says that “When $S \geq 20$, the performance achieves a comparable level” but there is almost a 3-4% increase when $S$ increases from 20 to 30 and also from 30 to 40. Then, it seems that increasing $S$ further should yield more improvements.
    * In any case, it would be good to identify when the performance starts saturating, to help users select the hyperparameter properly for other datasets.

* Missing discussion on training time and memory requirements
    * Given the extra prototypes involved, there should be some discussion on training time and GPU memory usage compared to the baseline DCC.

**Questions:**

* Please see the weaknesses section.

* Minor comments
    * Paragraph above Eq. 6 has typos, add space after citation of Gong et al. 2019.
    * Fig. 3: increase the font size of text inside the figure to match the caption font size. This will improve the readability of the plots.
    * Paragraph on “Effect of Loss” has a typo: “We” → “we”.
    * Table 6 (last row) has a typo: use math mode in LaTeX for $\mathcal{L}\_{cdd}$.

---

> ### Author Response · Authors · 2023-11-21
>
> Thank you for your recognition and suggestions on our work.
>
> **Re_Q1:** Effect of K
>
> Thanks for the advice. We conducted an additional experiment with **K = 1**, signifying the selection of only the most relevant item. However, this approach proved ineffective. The random initialization of the memory bank caused all input-oriented embeddings to converge towards selecting the same memory item. Additionally, in the visualization results, when **K =1**, the visualizations of memory items displayed meaningless images. The value of **K** is determined based on the attention values. During the design process, we observed that the fifth attention value is nearly zero. Consequently, we employ a top-K approach (**K = 5**) to filter out other noisy memory items.
> We will add these results analysis and visualizations to the revised manuscript.
>
> **Re_Q2:** Analysis of **S**
>
> Thanks for the advice. We conducted additional experiments with hyperparameters on the Office-Home dataset. When **S = 60**, the performance is **78.3%**. When **S = 80**, the performance is **77.5%**. These results suggest that **40** is the optimal **S** value for the Office-Home dataset considering **memory usage** and **performance**, and this optimal value may vary for other datasets with different degrees of intra-class concept shift. We will add this analysis to the revised manuscript.
>
> **Re_Q3:** Discussion on training time and memory requirements
>
> While DCC utilized approximately **6GB** of memory, MemSPM demonstrates efficient training on the Office-Home dataset using a **single RTX 3090 (using 10GB)**. The entire training process is completed within a day, establishing MemSPM as a computationally affordable and practical solution.
>
> **Minor comments:**
>
> Thanks for the advice. We have fixed these typos in the revised manuscript.

---

> > ### Comment · Reviewer_59Jy · 2023-11-23
> > **Response to authors**
> >
> > I thank the authors for their efforts during the discussion period. Most of my concerns have been addressed and I retain my already positive score of 8: accept, good paper.

---

### Meta-Review · Area_Chair_nx99 · 2023-12-05

**Metareview:**

R1, R2, and R4 liked the approach and experiments. However, R2 and R3 had concerns about some components of the method (i.e., sub-prototype and the method's reliance on DCC, a prior work). The rebuttal did not address the concerns. While AC agreed with R2 and R3's concerns, AC also found sufficient positive comments (e.g., novel approach, strong experiments) from the reviews to recommend the paper for acceptance.

**Justification For Why Not Higher Score:**

Two out of four reviewers were concerned about some components of the approach and the method's similarity to DCC, a prior work.

**Justification For Why Not Lower Score:**

Three reviewers liked the method and experiments.

---

### Decision · Program_Chairs · 2024-01-16

Accept (poster)